# An ECG Signal Acquisition and Analysis System Based on Machine Learning with Model Fusion

**DOI:** 10.3390/s23177643

**Published:** 2023-09-03

**Authors:** Shi Su, Zhihong Zhu, Shu Wan, Fangqing Sheng, Tianyi Xiong, Shanshan Shen, Yu Hou, Cuihong Liu, Yijin Li, Xiaolin Sun, Jie Huang

**Affiliations:** 1School of Aeronautical Engineering, Nanjing Vocational University of Industry Technology, Nanjing 210023, China; 2Innovative Research Laboratory of Nanjing Xi-Jing Advanced Materials Technology Ltd., Nanjing 211101, China; 3SEU-FEI Nano-Pico Center, Key Lab of MEMS of Ministry of Education, Collaborative Innovation, Center for Micro/Nano Fabrication, Device and System, Southeast University, Nanjing 210096, China; 4Key Laboratory of Optoelectronic Technology and Systems, Ministry of Education, Key Disciplines Laboratory of Novel Micro-Nano Devices and System Technology, School of Optoelectronics Engineering, Chongqing University, Chongqing 400044, China; 5School of Economics and Management, Nanjing Vocational University of Industry Technology, Nanjing 210023, China

**Keywords:** electrocardiogram, machine learning, model fusion, ECG, CNN

## Abstract

Recently, cardiovascular disease has become the leading cause of death worldwide. Abnormal heart rate signals are an important indicator of cardiovascular disease. At present, the ECG signal acquisition instruments on the market are not portable and manual analysis is applied in data processing, which cannot address the above problems. To solve these problems, this study proposes an ECG acquisition and analysis system based on machine learning. The ECG analysis system responsible for ECG signal classification includes two parts: data preprocessing and machine learning models. Multiple types of models were built for overall classification, and model fusion was conducted. Firstly, traditional models such as logistic regression, support vector machines, and XGBoost were employed, along with feature engineering that primarily included morphological features and wavelet coefficient features. Subsequently, deep learning models, including convolutional neural networks and long short-term memory networks, were introduced and utilized for model fusion classification. The system’s classification accuracy for ECG signals reached 99.13%. Future work will focus on optimizing the model and developing a more portable instrument that can be utilized in the field.

## 1. Introduction

According to the World Health Organization’s statistical report, the mortality rate for cardiovascular disease is significantly higher than that of any other disease globally. In 2012, this condition caused nearly 17.5 million deaths worldwide, representing approximately one-third of all fatalities. The China Cardiovascular Disease Report of 2018 reveals that regardless of whether individuals reside in urban or rural areas, they exhibit higher mortality rates related to cardiovascular disease than with other illnesses [1]. From 1990 to 2016, two out of every five deaths among rural residents in China resulted from cardiovascular disease. In this regard, data analysis indicates a mortality rate of 309.33 per 100,000 individuals due to cardiovascular conditions in rural areas, with such deaths accounting for 45.50% of total deaths. Comparatively, the death rate attributed to cardiovascular diseases in urban areas was 265.11 per 100,000 inhabitants, accounting for 43.16% of total deaths.

Upon overall analysis of the report, it is evident that the number of individuals with cardiovascular diseases in China had reached 290 million in 2017. Moreover, cardiovascular disease accounts for more than 40% of all causes of death, highlighting the significance of detecting such conditions. Within each normal beat of the heart lies a heart rate signal representing the organs’ normal functioning. This makes detecting heart rate signals an essential tool for preventing cardiovascular diseases. Currently, the primary focus of human health platforms involves designing wearable sensors that monitor biophysical, biochemical, and environmental signals. Implantable devices are also available for monitoring neurological, cardiovascular, digestive, and motor system activities [2]. Thus, the present research primarily emphasizes electrocardiography (ECG) as the most crucial tool for avoiding cardiovascular diseases.

ECG involves recording the electrical activity of the heart as a voltage–time graph via electrodes placed on the skin that detect myocardial depolarization during each heartbeat cycle. ECG primarily records the electrical activity of the heart and serves as a standard clinical cardiac test, providing an essential way to monitor cardiovascular disease while simultaneously examining for a variety of arrhythmias, ventricular hypertrophy, myocardial infarction, myocardial ischemia, and other abnormalities.

The heart consists of numerous cardiomyocytes, which are continuously electrically repolarized and depolarized. As the core unit of ECG, the heartbeat generates a variety of electrical stimulation pulses inside the heart that excite muscle cells in the atria and ventricles, resulting in regular diastole and contractile movements. Depolarization reflects electrical activity during contraction, whereas repolarization reveals the opposite activity. Following the beat cycle, the main elements of the heart, the atria and ventricles, undergo a contraction and diastole process that subsequently causes the heart to move. In this light, the ECG represents a voltage–time relationship of the electrical activity in the heart recorded by electrodes placed on the skin, resulting from a blend of physiological and technological processes. An electrocardiogram provides insight into the heart’s electrical activities, making it the most commonly used clinical heart test and a useful approach to monitor cardiovascular diseases while detecting any corresponding abnormal heart conditions [3].

As times change, the scale of data continues to expand considerably. Similarly, the collection of physiological signals from individuals is increasing dramatically, making manual data processing impossible [4]. Consequently, machine learning serves as a vital tool for data analysis. Machine learning enables the automatic extraction of data characteristics through simulating and analyzing portions of the data. It then provides a generalization model characterizing all data to facilitate processing massive datasets.

The creation process for a typical machine learning model involves collecting data to create a predictive model using training data. Some training data is filtered out and serves as verification data that is primarily used to identify the model. The accuracy of test data constitutes the prediction target. Machine learning extracts data features, enabling the model to learn from the training and the distinct features of data, subsequently predicting test data to conveniently acquire classification results. With regards to manual analysis, machine learning processes data rapidly; for instance, it takes less than 30 s to predict 10,000 heartbeat data points with a classification accuracy rate reaching up to 99%. However, training time is necessary in the early stages of testing, requiring at least 30 min to train 100,000 heartbeat data points. Therefore, machine learning proves crucial to enable efficient processing and categorization of big data compared with manual approaches. In the introduction, it is important to note that the heart rate signal, which represents the number of heartbeats per unit of time, can sometimes mistakenly be referred to as the BPM (beats per minute) signal. It is crucial to differentiate between the two to avoid any confusion or misinterpretation.

Improving the overall performance of ECG signal acquisition and analysis systems has become a critical research area in recent years. In this study, we address two core aspects to enhance the effectiveness of ECG signal analysis. Firstly, an embedded system is designed as the signal acquisition front end to improve portability. Secondly, we propose a machine-learning-based approach to tackle the problem of big data in ECG signal analysis. Specifically, we focus on the design of home-made printed circuit boards (PCBs) for ECG acquisition and the development of a reliable algorithm to analyze the acquired signals. Our ultimate goal is to create a complete system for ECG collection and AI-based analysis that allows for real-time monitoring of ECG signals outside of hospital settings [5]. This system can accurately detect abnormal heart rates and thus provide a robust tool for real-time health monitoring (Figure 1).

## 2. Materials and Methods

### 2.1. System Design

The ECG acquisition system is a critical component of the overall system functionality. Accurate identification of heart rate signals by signal analysis systems depends on the ability of the hardware to collect the correct signals. This, in turn, allows the machine learning model to accurately classify the data. The flow diagram of the full process is depicted in Figure 2.

The entire system can be divided into two parts: the acquisition front end and the analog-to-digital conversion section. Raw heart rate signals are collected via commercial electrodes using a triple-lead solution and transferred via wire to the acquisition front-end section for processing. After two stages of amplification and two digital filtering processes, the ECG signal is finally transmitted through an amplifier for analog-to-digital conversion and processed through a digital filter. In the final stage, the acquisition system delivers heart rate ECG signals suitable for subsequent classification.

To address the issue of environmental noise, which can lead to amplitudes much larger than the collected ECG signal, appropriate amplifiers should be chosen based on the characteristics of the noise. Useful ECG signals typically lie between the frequencies of 0.05 Hz and 100 Hz, necessitating optimized filters for interference cancellation. Following the amplifier and filter stages, the signal requires sampling and analog-to-digital conversion. According to the Nyquist sampling theorem, the sampling frequency must be at least twice that of the original signal to ensure its integrity [6]. A sampling frequency of 200 Hz is sufficient to prepare the signal for digital processing. However, since the spectrum of some diseases may extend as high as 500 Hz, the minimum sampling frequency during system design should be 1000 Hz. The choice of sampling frequency depends on the complexity of the analysis of the ECG signal. For remote signal monitoring, wireless communication standard modules such as Zigbee, RFID, or Wi-Fi are necessary [7]. The collected data are then transmitted to a terminal where they are received, stored, and finally processed.

### 2.2. Circuit Design

Considering the fundamental requirements of the acquisition front end, selecting an appropriate front-end circuit chip is pivotal from various perspectives. To address potential external interference in distributed devices, an integrated dedicated chip for heart rate signal acquisition with built-in amplification and filtering circuits will be utilized. These components are highly integrated and resistant to external interference. Currently, the most widely used ECG signal acquisition chip on the market is the AD8232. This chip presents several advantages such as the option to choose between two- or three-lead modes for transmission.

Its signal gain (G) can reach up to 100, whereas its high-pass filter adopts a double-pole structure and has a fast recovery function, thereby reducing the long tail phenomenon caused by the high-pass filter of the two-pole structure. Additionally, the module features lead-off detection with an integrated right-leg-drive (RLD) amplifier, which improves the common-mode rejection ratio (CMRR). However, this type of structure may have insufficient accuracy and the package volume of the chip is relatively large. Integrated designs have many disadvantages compared with distributed designs, potentially amplifying the effect of external interference factors on signal collection and leading to increased noise in the final analog signal that could significantly affect subsequent signal analysis and classification [8]. After achieving a balance between speed and performance, AD8232 was chosen as the optimal chip for this system, combining built-in primary and secondary amplification while working together with peripheral circuits to achieve both high-pass and low-pass filtering.

Upon completion of data collection, it is essential to utilize the three-lead method to increase signal strength and reduce interference from artifacts. This approach also minimizes the impact of common-mode interference on the collected heart rate signal via the RLD in AD8232, which is also utilized for driving the third electrode. During signal collection and transmission, two 180 kΩ resistors are introduced to connect to the input pin as a protective measure against faulty input signals, whereas two 10 MΩ resistors are implemented for input bias. The core task involves implementing an integrated acquisition front-end circuit design using the AD8232 chip in combination with peripheral circuits. Such an approach aims to reduce the design area and improve system portability and stability while simultaneously enhancing the quality of the collected signals, together with the post-processing accuracy and overall diagnostic system confidence. A schematic of the final circuit design is illustrated in Figure 3.

The primary objective of the acquisition front-end circuit is to obtain as many original heart rate signals as possible. In contrast, the digital part of the circuit focuses on receiving analog signals, performing analog-to-digital conversion, and ultimately communicating with external devices via Bluetooth modules. The circuit primarily comprises the main control chip and peripheral circuits. The data transmission procedure involves a digital circuit that receives the heart rate analog signal from the acquisition front end. Subsequently, other circuits perform analog-to-digital conversion, digital filtering, and ultimately transmit the processed data through the Bluetooth module. The critical modules that require designing are the MCU (microcontroller unit) selection, peripheral circuit design, power supply module design, and Bluetooth module design. It is crucial to ensure the system’s stability and power consumption during the design process.

### 2.3. Circuit Realization

Altium Designer 20 was utilized to design the printed circuit boards (PCB) in this study. Figure 4a,b depict the front-side and back-side schematic diagrams of the acquisition front-end PCB circuit.

Meanwhile, Figure 4c,d present the PCB circuit diagram of the ECG acquisition system prior to and after installation of the Bluetooth telecommunication module, respectively. Following installation, the entire signal acquisition system can be visualized in Figure 4e, which exhibits the analog signal output from the acquisition front end through an oscilloscope. Through careful inspection, we verify that the acquisition front end can effectively collect a diverse range of abnormal heart rate signals.

The oscilloscope employed in this work is the Agilent DSO7104B. The diagram of normal heart rate, parameter 300 beats per minute (BPM), with a frequency of 5 Hz. Figure 5A shows atrial tachycardia, which is generally manifested in the atrium. The rate was usually 150–200 beats/min, and the P-wave morphology was different from that of sinus patients. By anterior measurements, it was evident that the P-wave pattern was abnormal and the atrial rate was rapid. Figure 5B shows supraventricular tachycardia, which is one of the most common arrhythmias. Its clinical manifestation is a series of rapid, regular QRS waveforms with a frequency of 160 to 220 BPM. Figure 5C is atrial fibrillation; referred to as atrial fibrillation, it is the simplest persistent arrhythmia with a frequency of 300–600 BPM, a rapid heart rate. The signals from the front end of the acquisition can indicate that atrial fibrillation does occur with rapid heart rate. Figure 5D shows ventricular tachycardia, which refers to tachyarrhythmias occurring in bundle branches, myocardial conduction fibers, and ventricular muscles below the bifurcation of the bundle. Higher frequencies and abnormal QRS waves can be accurately detected through the acquisition front end through Figure 5E–L, as shown.

Once the data signal is collected by the acquisition front end, which operates at a sampling rate of 360 Hz, the analog signal is transmitted to another module via HC-05 Bluetooth communication technology. In this study, serial port communication is utilized to connect with the terminal system. Subsequently, the recorded data is stored on the terminal system for further processing. Our experimental outcomes validate that the terminal can effectively transmit and record ECG data.

### 2.4. Schematic System Design

The ECG analysis system represents the ultimate outcome of the overall plan. It is fundamental to accurately collect heart rate signals, as their proper classification facilitates the proper functioning of the entire system. The ECG signal received by the terminal is usually subject to noise. Thus, the signal requires preprocessing. Noise removal should be primarily considered. Additionally, ECG signal classification typically relies on heartbeats. Therefore, segmenting ECG signals becomes necessary. Since QRS waveforms are more recognizable, identifying QRS waveforms first and selecting fixed signal points on both sides serves as an idea for heartbeat segmentation. After this segmentation, it is essential to divide the data into the training set and test set, construct feature engineering, perform traditional model and deep model training individually, and combine the prediction results to obtain the final classification outcome. The design flow chart of the ECG analysis system is presented in Figure 6.

### 2.5. Software Design Process

In this study, the internationally recognized standard ECG database has been utilized, with specific emphasis on the MIT-BIH database. This database consists of 48 ECG records, each of which is 30 min in length, sampled at a rate of 360 Hz, and contains a total of 648,000 sampling points. It should be noted that actual training data comprised mainly sampling points corresponding to each heartbeat that were generally set between 200–300 points; for this experiment, 250 points were selected as a compromise. To address the four classification problems observed in the treatment phase, three types of abnormal ECG signals, including left bundle branch block, right bundle branch block, and ventricular premature beats, were used. ECG signals are bioelectrical signals and are vulnerable to external interference, such as common electromyography interference and power frequency interference. Therefore, despite the design of filters in both the acquisition front end and digital circuits, the original data still required preprocessing before being fed into the software analysis to enhance the signal-to-noise ratio of the machine learning model training dataset. The primary algorithm employed for this purpose was wavelet transform.

## 3. Results

### 3.1. Data Preprocessing

The wavelet transform denoising methodology functions through the decomposition of signals into a superposition of wavelet functions. Following wavelet decomposition, wavelet coefficients are generated for each signal. Subsequently, the signal and noise are distinguished using threshold settings, in which a suitable threshold allows for larger signal coefficients and smaller noise coefficients. As such, the test signal is focused on specific frequency bands, whereas noise is dispersed across the entire frequency spectrum. Gaussian white noise performance can be identified by setting the appropriate threshold, even in the wavelet domain, given that its performance is random. Throughout the experiment, three common wavelet basis functions (bio2.6, db8, and sys8) were implemented and tested separately to achieve optimal results. Figure 7 provides a clear visualization of the denoising effect, where Figure 7a represents the original signal. The essence of wavelet transform is essentially the same as that of Fourier transform, both of which use a basis to represent signals. The difference lies in the fact that wavelet transform uses a set of bases, where ‘wavelet’ refers to a type of wave with energy highly concentrated in the time domain, with limited energy and concentration at a specific point. Simultaneously, it has the ability to characterize local features in both the time and frequency domains [9,10,11]. Compared with Fourier transform, the advantage of wavelet transform is its ability to handle abrupt or discontinuous signals, and it can also perform refined analysis of signals through stretching and shifting operations [12].

It is evident that the wavelet-denoised signal exhibits reduced noise compared with the original signal. Generally, the assessment of signal quality involves analyzing the signal-to-noise ratio while also taking into account the need to minimize excessive distortion of the overall signal during denoising. Accordingly, the denoised signals must be compared with their corresponding original signals to detect any inconsistencies [13]. Methodologically, mean absolute error (MAE) and root mean squared error (RMSE) are two commonly used indicators for measuring variable accuracy [14]. In this study, MAE was employed in conjunction with the signal-to-noise ratios to determine bior2.6 as the optimal wavelet basis function.

### 3.2. QRS Waveform Positioning

As the previous processing was based on slope extraction or low-pass filtering, there may be unstable amplitudes and a decreasing trend. In order to ensure the continuity of the overall curve, the idea of exponential weighted averaging that is commonly applied in machine learning is considered. Therefore, a sliding integration method is introduced into this model, which smooths the curve through weighted averaging. The length of the window utilized in operation is also an empirical parameter that is generally set between 10–20 and then fine-tuned. Through the sliding integration method, the amplitude can be increased significantly and the signal is composed of wave peaks, which facilitates detection. The subsequent task is to detect the wave peaks, with each wave peak corresponding to a QRS wave.

Due to the fact that the QRS wave is composed of double-sided peaks, there will be a phenomenon of bimodality in the actual algorithm implementation. Although the waveform is singular, it may reduce the detection accuracy. Therefore, a low-pass filter is used to smooth the waveform. The setting of the cutoff frequency is an empirical parameter, generally between 5 Hz and 10 Hz, that can effectively filter out the noise without affecting the subsequent operations.

Finally, the threshold is set. In this paper, an adaptive threshold using a sliding average method is applied, meaning that a threshold is set for each wave peak and that if the amplitude is greater than the threshold, then the position is identified as a wave peak or a QRS wave. The threshold is set as the mean of past peaks multiplied by a coefficient. In the paper using double slope, a simple coefficient of *H*_cur_ > *H*_ave_ × 0.4 is adopted as the threshold, where the parameter *H*_cur_ represents the slope of the current heartbeat and *H*_ave_ refers to the mean slope of the past eight wave peaks; this method is used in order to ensure the overall stability of the signal during the practical applications. The parameters of the threshold and some basic constraints are set up in order to form an adaptive threshold method, as follows.
Calculate *θ*_0_:*θ*_0_ = *H*_ave_ × 0.65,if *P* > *θ*_0_;*θ*_0_ = *θ*_0_ − (*P* − *H*_ave_) × 0.5,if *θ*_1_ < *P* < *θ*_0_;*θ*_0_lim_,if *θ*_0_ < *θ*_0_lim_;Calculate *θ*_1_:*H*_ave_ × 0.35,if *P* > *θ*_0_;0.35 × *P*,if *θ*_1_ < *P* < *θ*_0_;*θ*_1_lim_,if *θ*_1_ < *θ*_1_lim_;

In this method, *θ*_0_ and *θ*_1_ are the adaptive high and low thresholds, respectively, that ensure the stability of the threshold. *P* represents the current peak value, whereas *θ*_0_lim_ and *θ*_1_ are artificially set parameters to prevent the threshold from being set inappropriately. The purpose of this adaptive threshold method is to ensure that signals are not missed. If the signal is greater than *θ*_1_, it indicates a wave peak, and *θ*_0_ is dynamically adjusted. During adjustment, if the wave peak is higher than *θ*_0_, it means that the threshold is too small. Thus, *θ*_0_ and *θ*_1_ are required to be increased simultaneously and the coefficients need to be adjusted accordingly. Otherwise, they should remain stable. Figure 8 represents the results.

The waveform is identified as a feature and consists of various forms, including the QRS wave, *P*-wave, and *T*-wave. Therefore, it is essential to locate each waveform type before subsequent analysis. However, in practical problem-solving, identifying actual heartbeats is challenging since the acquisition front end only collects voltage signals that necessitate heartbeat identification. In research contexts, the QRS wave has typically been employed for heartbeat positioning, given that this wave displays a sudden change in slope featuring a peak that more easily detects the waveform. Furthermore, in practice, the QRS wave is observable, making it the easiest waveform to identify during monitoring. The current commonly used Pan–Tompkins algorithm mainly utilizes a bandpass filter to achieve differentiation, squaring, and integration (herein lies the important implementation for slope processing), followed by an adaptive threshold search process [15]. Although it has relatively excellent performance, it is complex to implement. It is possible to simplify this section while still using the basic double-slope idea.

By capitalizing on the significant slope at both ends of the QRS waveform, the dual-slope method is employed to extract the QRS wave. Essentially, this method uses the slopes on either side of the peak to determine its location. Three primary criteria are considered: the shape, the height, and the steepness (slope) of the signal. The maximum and minimum slopes on each side are denoted as *S*_L,max_, *S*_R,max_, *S*_L,min_, and *S*_R,min_, respectively. Generally, the duration of the QRS waveform from peak to baseline falls within the range 0.03 s–0.05 s. Therefore, when processing a sample, signals within the range 0.03 s–0.05 s of the subsequent sample are mainly processed [16]. Afterward, the maximum values for the slope on both sides are computed and used to calculate the maximum value on the left minus the minimum value on the right:*S*_diff,max_ = MAX((*S*_L,max_ − *S*_R,min_),(*S*_R,max_ − *S*_L,min_))

Similarly, the maximum on the right minus the minimum on the left is computed. The maximum value between the two differences is then selected and compared against the threshold. If the difference exceeds the threshold, it indicates a QRS waveform signal. The flowchart provided in Figure 9 depicts the process visually.

### 3.3. Deep Learning Model

Feature engineering is crucial for traditional machine learning models since they require solid interpretability [17]. In the case of ECG signals, the design space for feature engineering is limited, making it challenging to achieve ideal results, particularly with respect to QRS and *P*-wave extraction. Therefore, considering a deep learning approach becomes necessary. Convolutional neural networks (CNN) are ideally suited for image processing [18]. After thorough analysis, we observe that ECG signals share similar traits with images, including that local waveform details significantly affect the entire signal. Consequently, employing a neural network allows for automatic feature extraction while bypassing the manual feature engineering and waveform extraction steps. Furthermore, long short-term memory (LSTM) networks, consisting of time loops, are commonly applied in sequence data processing. ECG signals are, in essence, a typical heartbeat sequence. We propose using two such networks to develop a classification model, subsequently merging them to analyze our findings.

First introduce the overall structure of the CNN network as shown in Figure 10. The actual structure consists of four convolutional layers, two pooling layers, and four fully connected layers. The size of the first convolution kernel is 11 × 1 and then the dropout layer is connected. The reason for the 3 × 1 size of the following three convolution kernels is mainly because the adjacent information of the ECG signal has a greater impact, so the use is more. The small convolution kernel ensures that information is not leaked as much as possible.

### 3.4. LSTM Model

LSTM (long short-term memory) is an improved model based on recurrent neural networks (RNNs). The improved method introduces a gate mechanism to solve the problem of gradient disappearance and gradient explosion in RNNs. The unit structure diagram of LSTM is shown in the figure, where the input of the unit includes the current sequence input *x*_t_, the state of the hidden layer at the previous moment *C*_t−1_, and the output vector *h*_t−1_ and the input of the unit is the current hidden state *C*_t_ of the layer and the output vector *h*_t_. The unit contains three gates. The forget gate determines the degree of retention of the previous cell state *C*_t−1_, the input gate determines the information transmitted to the model by the current sequence input *x*_t_, and the output gate determines the final neuron, the current state *C*_t_ and output vector *h*_t_ [19]. The overall architecture diagram is shown in Figure 11.

The core of LSTM is the status of the cell, as represented in Figure 12. The algorithm of the forget gate is to selectively retain the status of last cell *C*_t−1_, as shown:*f*_t_ = *σ* (*W*_f_ [*h*_t−1_, *x*_t_] + *b*_f_)while *f*_t_ is the output factor of the forget gate, *W*_f_ and *b*_f_ are the weights and biases to be trained. [*h*_t−1_, *x*_t_] represents the combined vector of the output value *h*_t−1_ from the previous unit and the input *x*_t_ of the current sequence [21].

The forget gate selectively maintains information by retaining or discarding it based on the combination of these inputs. *σ* refers to the sigmoid activation function [22]. The input gate is utilized to extract the current sequence input and the input value from the previous layer unit and calculate the input factor [23].*i*_t_ = *σ* (*W*_i_ [*h*_t−1_, *x*_t_] + *b*_i_)
*Ĉ*_t_ = *tanh*(*W*_c_ [*h*_t−1_, *x*_t_] + *b*_c_)while the parameter *i*_t_ is the retaining parameter, *Ĉ*_t_ is the value of input gate. *W*_i_ and *W*_c_ are the weight and *b*_i_ and *b*_c_ are the biases [24].

### 3.5. Model Fusion

The basic principle of model fusion to improve the final classification results can be likened to Condorcet’s jury theorem. According to this theorem, if each juror’s decision is independent and the probability of accurate judgment is higher than 50%, then the combined result of all jurors tends towards certainty. From a machine learning perspective, each juror corresponds to a different model, regardless of the strength of the model [25]. Their accuracy must be higher than 50% (where 50% represents random guessing). Therefore, the weaker the correlation between models, the better the effect of model fusion [26].

There are two main fusion methods commonly utilized in practice:The voting method, which can be further divided into soft voting and hard voting. Hard voting involves selecting the most frequent classification result among all models for a classification problem, whereas soft voting involves taking the average of the predicted probabilities from all models.The stacking method, which utilizes a layered training approach. It begins by dividing the training set into five subsets. Model 1 is trained using five-fold cross-validation, where each time it is trained on four subsets and predicts the remaining subset as well as the test set. This process is repeated five times with different combinations of four subsets. The predictions from these five iterations are then concatenated to form new features and labels for the training set. The average of the five predictions on the test set is also generated as new features for the test set. Model 2 is trained using these new features and is used to obtain the final results for the test set. Stacking has the advantage of utilizing multiple rounds of cross-validation, providing stronger stability and reducing the risk of overfitting. Therefore, in practical applications, stacking is often chosen as the model fusion method [27]. The flowchart of the stacking method is illustrated in Figure 13.

### 3.6. Software Realization

The whole diagram of classification algorithm is shown in Figure 14. First of all, the original ECG signal is denoised by wavelet. Then, the QRS wave recognition is performed and the heartbeat is captured. Then, the traditional model and the depth model are designed. The traditional model needs to design feature engineering, and the depth model needs to design model structure. The final result is obtained by model fusion after training and parameter adjustment.

## 4. Discussion

Multiple types of models were built for overall classification and model fusion was conducted. Firstly, traditional models such as logistic regression (LR) [27], support vector machines (SVMs) [28], and XGBoost were employed, along with feature engineering for classification [17]. Feature engineering primarily included morphological features and wavelet coefficient features. Subsequently, deep learning models, including CNN and LSTM, were used for classification. The classification results of LR and the SVM are shown in Table 1.

The experimental results of LR, the SVM, and XGBoost were analyzed (Table 2 and Table 3). The overall performance was improved by combining wavelet transformation for coefficient extraction with meaningful first- and second-level features. Additionally, PCA and LDA were employed to reduce the dimensionality of features, ensuring that the feature space was not excessively large. The overall classification accuracy exceeded 96%, with a higher accuracy for normal beats.

However, there was a lower classification accuracy for ventricular premature beats. The core features of ventricular premature beats include a relatively wide and deformed QRS complex, ST segment, as well as a T-wave direction opposite to the QRS complex. The QRS duration is greater than or equal to 0.12 s (the normal limit is less than or equal to 0.11 s). Analysis revealed that the main reason for the lower accuracy was that manual feature engineering fails to effectively capture the distinctive features of ventricular premature beats, especially the differentiation between ST segment and *T*-wave direction and QRS complex direction. Furthermore, the discriminatory power of QRS duration is insufficient, which is reflected in the relatively low feature importance obtained from the XGBoost classifier. Therefore, it is necessary to complement the model using deep learning techniques [29].

XGBoost has several parameters to adjust, as shown in Table 2. The main parameters that need to be adjusted include:Learning_rate (eta): This represents the learning rate or step size for the iteration of decision trees. A smaller value leads to a slower iteration, whereas a larger value may result in lower accuracy.Max_depth: This parameter determines the maximum depth of each decision tree. It is generally set between 5 and 10. Setting it too high can lead to overfitting.Random seed: The random seed can be adjusted using cross-validation to ensure reproducibility and control randomness.colsample_bytree and colsample_bylevel: These parameters control the proportion of randomly sampled rows and columns, respectively. They are used to mitigate overfitting by introducing randomness into the sampling process.To select the best parameters, it is recommended to perform testing and cross-validation. The parameter settings of CNN are shown in Table 3.

An analysis was conducted on the experimental results of CNN and LSTM, revealing an improvement of approximately 1.4% compared with traditional models and feature engineering [30]. Furthermore, the performance in detecting ventricular premature beats (VPBs) was better than that of the traditional models. This suggests that the previous approaches of wavelet transformation and manual feature construction were not able to effectively identify the distinguishing characteristics of VPBs from other signals. Therefore, it is necessary to explore various aspects of feature engineering. In the case of large datasets, when proper parameters are set, deep learning generally outperforms traditional machine learning methods [31]. Additionally, the final experimental results of model fusion are shown in Table 4 below.

To validate the accuracy of the overall model’s predictive results in practical measurements, two sets of data were collected comprising five left bundle branch block heartbeats and five right bundle branch block heartbeats. The actual prediction accuracy was found to be 100%. It is evident that the accuracy of the model’s predictions in practical scenarios is exceptionally high.

The classification of V (ventricular premature beats) is relatively challenging compared with other symptoms in the overall context. The main reason for this difficulty lies in the inadequate depiction of distinguishing features related to the ST segment, T wave direction, and QRS wave direction. Consequently, the model fails to recognize these features effectively, indicating a need for special design considerations in characterizing the features and model structure for ventricular premature beats. Meanwhile, it is observed that deep models outperform traditional models in general, suggesting that the wavelet coefficients and morphological features do not sufficiently allow the model to learn the differences between ventricular premature beats and other states. Thus, more effective features in this regard should be explored.

Furthermore, within the overall context, except for the CNN’s superior performance in classifying R (right bundle branch block), the accuracy of the fused models is better for the remaining cases. This highlights that fusion of models with low correlation indeed enhance the final predictive results effectively.

Lastly, the accuracy of the model in predicting the normal state is nearly similar. This implies that the manually constructed feature set can effectively depict the heart rate during the normal state, resulting in the highest overall accuracy in normal state classification.

In order to ensure the standardization of the research, it is necessary to adjust the classification criteria according to the AAMI (Association for the Advancement of Medical Instrumentation) standards. AAMI and MIT-BIH fundamentally represent two different classification methods in clinical practice.

Therefore, in order to meet the standards set by AAMI, the classification approach of the AAMI was employed to transform the problem into a five-class classification task. However, due to limited training data for the Q category in the MIT-BIH dataset, this category was excluded, resulting in a four-class classification problem for practical application. To expedite the training process, an improved CNN model was utilized for prediction, as depicted in Figure 10. In comparison with the previous CNN model, the core component of the modified model involved adjusting the size of the intermediate convolutional kernels from 31 to 91, 61, and 31. The reason for not using larger kernel sizes was because heartbeats inherently consist of short-term data and CNN models are better suited to extracting features by starting with smaller-sized kernels [32]. The use of different kernel sizes aims to maintain small sizes while maximizing the window for feature extraction. The final accuracy achieved was 99.16%, as indicated by the actual test results presented in Table 5.

The analysis of the final results leads to the following conclusions. It can be observed that there is an issue of sample imbalance in the S, V, and F classes within the MIT-BIH dataset. Therefore, in the initial phase of the study, these classes were intentionally avoided, and only the N class was used for classification. This approach indeed helped to address the problem of sample imbalance to a certain extent. To tackle sample imbalance, three common strategies are generally employed:Weighted processing—where different weights are assigned to different samples in the final loss function;Oversampling—which involves artificially increasing the number of samples by applying a sampling multiplier to the minority class;Undersampling—which reduces the number of samples in the majority class [33].

In practice, all three methods were attempted. However, due to the scarcity of samples in certain categories, for example, the F class only had 800 samples in the training set, undersampling was not deemed suitable in some cases, whereas oversampling did not effectively improve the training performance [34]. Therefore, the focus should be on addressing the imbalance issue at the data level.

The majority of the training data for this study is sourced from the MIT-BIH database. Due to the limited volume of data collected by the embedded system, only a small sample test could be conducted. Therefore, the accuracy achieved does not fully represent the accuracy of this classification system in handling real-time heart rate data. As a result, further in-depth research is required to investigate the real-time performance of the system [35].

This chapter focuses on the software component of the entire ECG analysis system. Whereas the hardware component is responsible for signal acquisition, the software part is responsible for signal analysis and classification. Firstly, the overall machine learning classification process is defined, including data preprocessing, dataset partitioning, feature engineering, model selection, performance testing, and model fusion. In the data preprocessing phase, attention is given to addressing certain issues in heart rate signals by applying wavelet transformation for denoising purposes. Due to the specific nature of heart rate signals and to facilitate feature engineering, a QRS wave detection algorithm was designed. Subsequently, the dataset was partitioned and three partitioning methods were proposed and selected based on theoretical and practical considerations. Feature engineering was constructed from two perspectives: morphological features and frequency domain features. For model selection, traditional models such as LR, SVMs, and XGBOOST were initially tried in combination with manual feature engineering. However, considering the limitations of traditional models and the strong fitting capabilities and non-requirement of manual feature engineering in deep learning models, LSTM and CNN classic models were adopted for classification. Additionally, traditional models were used for model fusion to prevent overfitting [36]. Finally, starting from the AAMI standard, a new model was constructed with new classification criteria and the results were analyzed.

## 5. Conclusions

The main research content of this paper is the study of an ECG signal acquisition and analysis system based on machine learning, and the main achievement is a system that can collect ECG signals and transmit them to the terminal for classification. The overall framework of the system is mainly an ECG signal acquisition system and an ECG analysis system. The signal acquisition system is responsible for collecting ECG data from the human body and sending it to the terminal. The analysis system mainly includes data preprocessing and machine learning models. Software modules are written on the terminal for data reception. The data is preprocessed and then a machine learning model is constructed to classify and set evaluation indicators to evaluate the model results, which mainly improves the convenience of the ECG acquisition system and realizes a system for detecting whether the human heart rate signal is normal. The final experimental results and conclusions are as follows:

(1) Use self-made PCB to realize the design of ECG acquisition system and improve the portability of the system. The overall hardware design mainly includes signal transmission and processing. The main structure of the acquisition system is to first collect ECG signals through commercial electrodes and three-lead methods (the main reason for using three-leads is that three-leads are more accurate than single-leads and are simpler and more portable than twelve-leads). Then, use AD8232 as the acquisition front-end chip. The built-in circuit of the AD8232 chip has the functions of primary and secondary amplification and high-pass and low-pass filtering. Then, use STM32F103 as the main control chip to collect the analog signal obtained by AD8232 through the ADC module after the signal is digitally filtered. Finally, the converted digital signal is sent to the terminal. Finally, it is concluded that a circuit with AD8232 and STM32F103 as the core can accurately collect most types of heart rate signals (normal heart rate and multiple abnormal heart rates).

(2) Design a machine learning model to analyze the collected ECG signals and combine the embedded system to realize a complete system for ECG collection and artificial intelligence analysis. The overall scheme design of the analysis system mainly includes the preprocessing of the ECG signal and the idea of model construction. Collect the data through the terminal and then use the wavelet transform denoising method to filter the data; after the filtering is completed, use the QRS extraction method to achieve the extraction of the heartbeat. The core idea is to use the dual-slope algorithm to extract the position of the QRS wave. The position of the QRS wave divides the data points forward and backward to realize the heartbeat segmentation. The length of the data points is generally in the range of 200–300. In this article, we use 100 points forward and 150 points backward, totaling the division of 250 points that are used as the heartbeat length. We then build different types of models for an overall classification and model fusion. First use traditional models (including logistic regression, support vector machine, XGBOOST) and feature engineering for classification, feature engineering mainly includes morphology features and wavelet coefficient features, then use deep learning models (including convolutional neural networks and long- and short-term memory networks) for classification. Finally, all models are fused to obtain the final result, and the final accuracy of the model is 99.13%.

To deploy the proposed system for real-time monitoring of ECG signals and analysis of potential heart diseases, there are different approaches for implementation in hospitals or at home:

Hospitals: In a hospital setting, the system can be integrated into the existing infrastructure for cardiac monitoring. The ECG monitoring device can be connected to a central monitoring station through wired or wireless connections. The ECG signals can be transmitted in real-time to a dedicated server, where the AL machine learning algorithms are deployed for analysis. The results can be displayed on a monitoring dashboard accessible to healthcare professionals, allowing them to make timely diagnoses and decisions.

Home use: For deployment in a home setting, a portable ECG monitoring device can be developed. This device should be user-friendly, compact, and able to wirelessly transmit the ECG signals to a mobile app or a cloud-based platform. The app or platform can then apply the AL machine learning algorithms to analyze the signals and provide feedback on potential heart diseases. Users can securely access their ECG data and analysis results through the app and, if necessary, share them with healthcare professionals for remote monitoring and consultation.

In both scenarios, it is important to ensure the system’s accuracy, reliability, and data security. Compliance with the relevant regulatory standards and obtaining the necessary approvals may also be required before deploying the system, especially in a medical context.

## Figures and Tables

**Figure 1 sensors-23-07643-f001:**
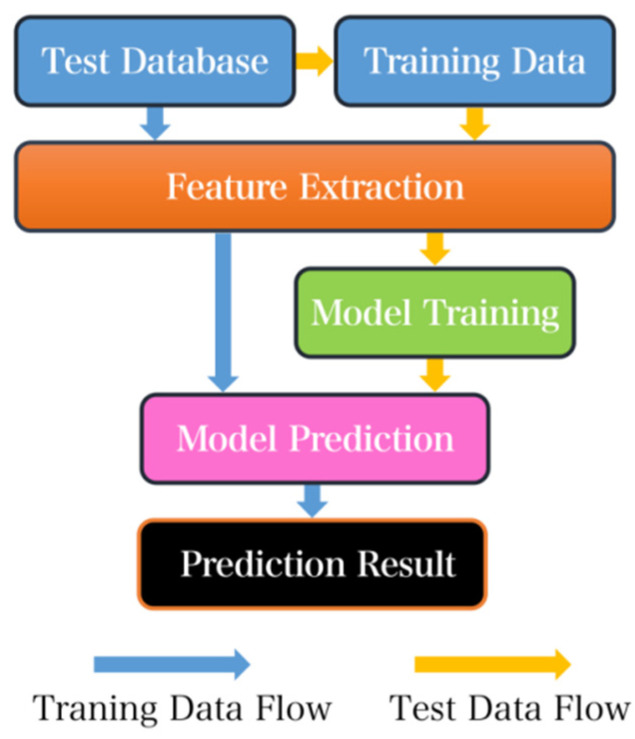
Common machine learning model building process diagram.

**Figure 2 sensors-23-07643-f002:**
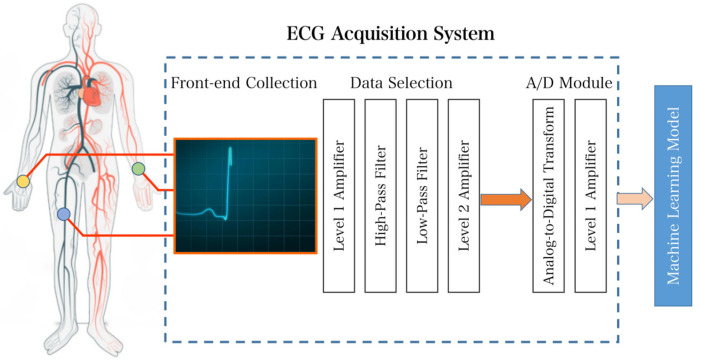
Overall system design flow chart of the ECG acquisition system.

**Figure 3 sensors-23-07643-f003:**
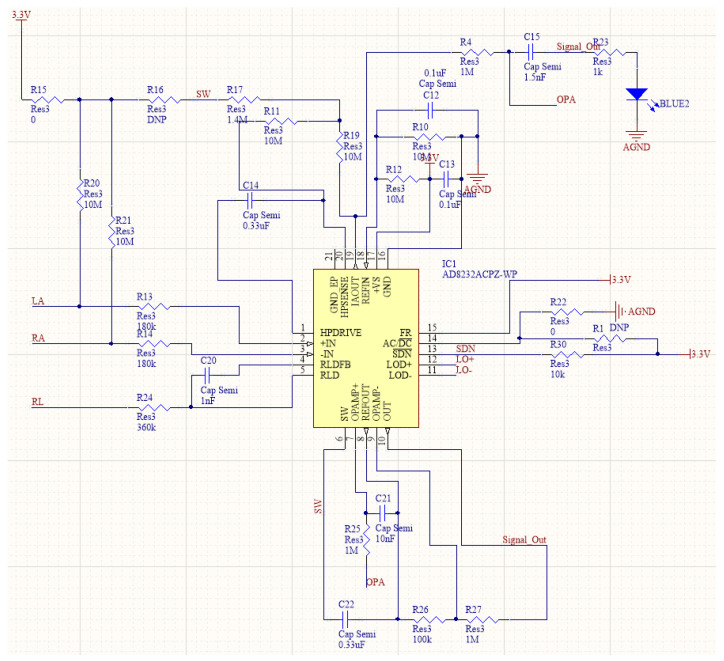
Design of acquisition front-end overall circuit.

**Figure 4 sensors-23-07643-f004:**
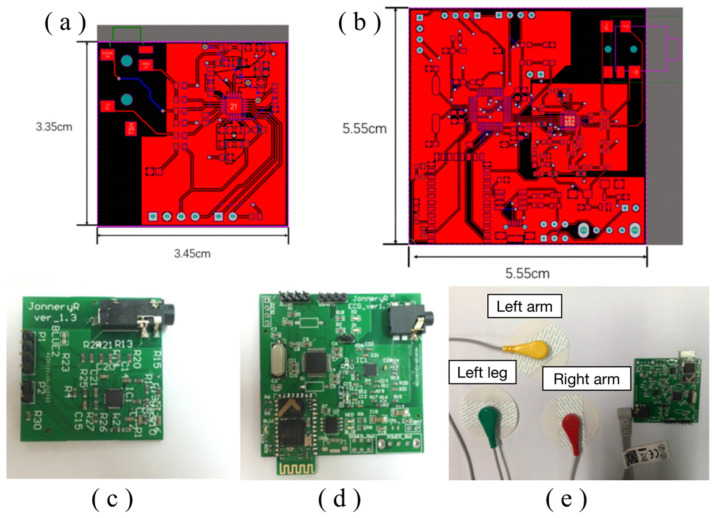
PCB circuit diagram of (**a**) front end of acquisition; (**b**) whole embedded system; and fabricated PCB of (**c**) front end of acquisition; (**d**) whole embedded system; and (**e**) 3-lead electrocardiogram and electrodes connected to the fabricated PCB device.

**Figure 5 sensors-23-07643-f005:**
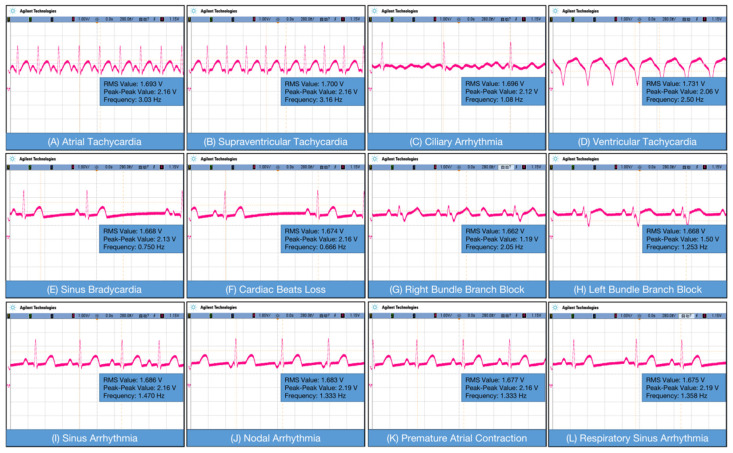
Test ECG signal of: (**A**) atrial tachycardia; (**B**) supraventricular tachycardia; (**C**) ciliary Arrhythmia; (**D**) ventricular tachycardia; (**E**) sinus bradycardia; (**F**) cardiac beat loss; (**G**) right bundle branch block; (**H**) left bundle branch block; (**I**) sinus arrhythmia; (**J**) nodal arrhythmia; (**K**) premature atrial contraction; and (**L**) respiratory sinus arrhythmia.

**Figure 6 sensors-23-07643-f006:**
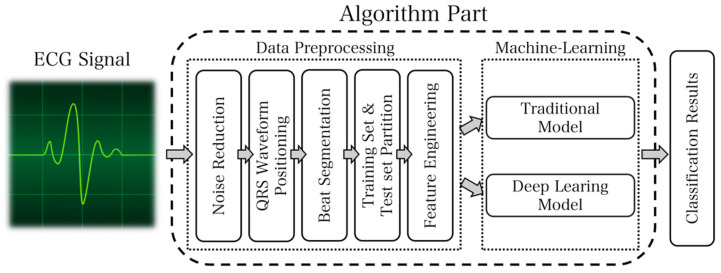
The algorithm part of the data processing module and machine learning module.

**Figure 7 sensors-23-07643-f007:**
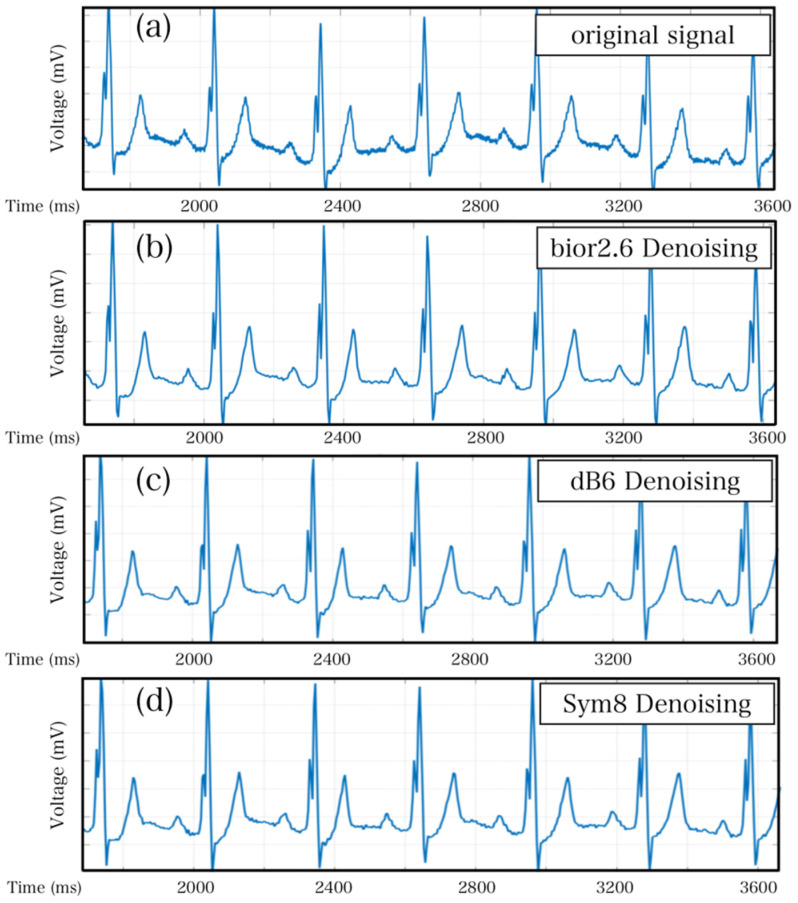
Wavelet denoising effect diagram. (**a**) The original signal before denoising. (**b**) The denoising effect after using the wavelet basis function bior2.6. (**c**) The denoising effect after using the wavelet basis function db6. (**d**) The denoising effect after using the wavelet basis function sym8.

**Figure 8 sensors-23-07643-f008:**
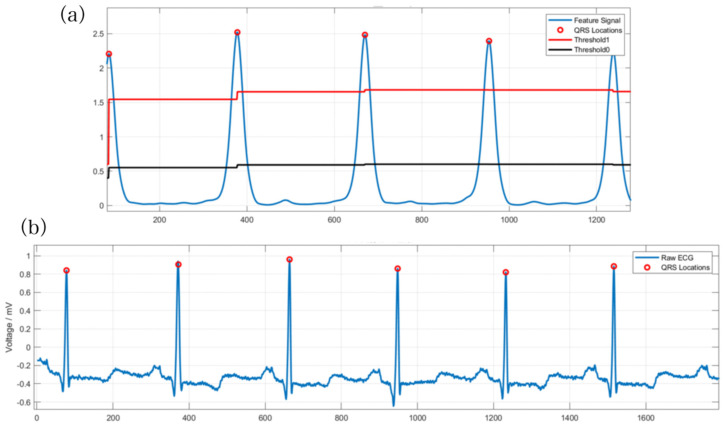
QRS waveform localization experiment result, the red circle marks the peak positions of the detected QRS waves. (**a**) The effect of QRS wave localization after filtering. (**b**) The experimental results of identifying QRS waves.

**Figure 9 sensors-23-07643-f009:**
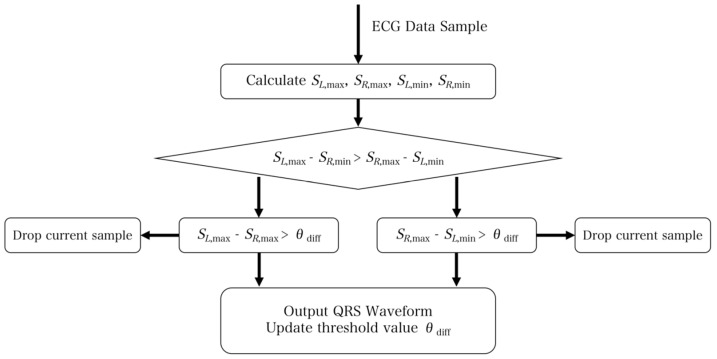
Flow chart of the double-slope rough positioning QRS waveform. The core algorithm is to calculate the maximum value from the left and right slope, then compare each other after subtraction and set the threshold. If the value is higher than the threshold, it indicates that a QRS waveform is detected.

**Figure 10 sensors-23-07643-f010:**
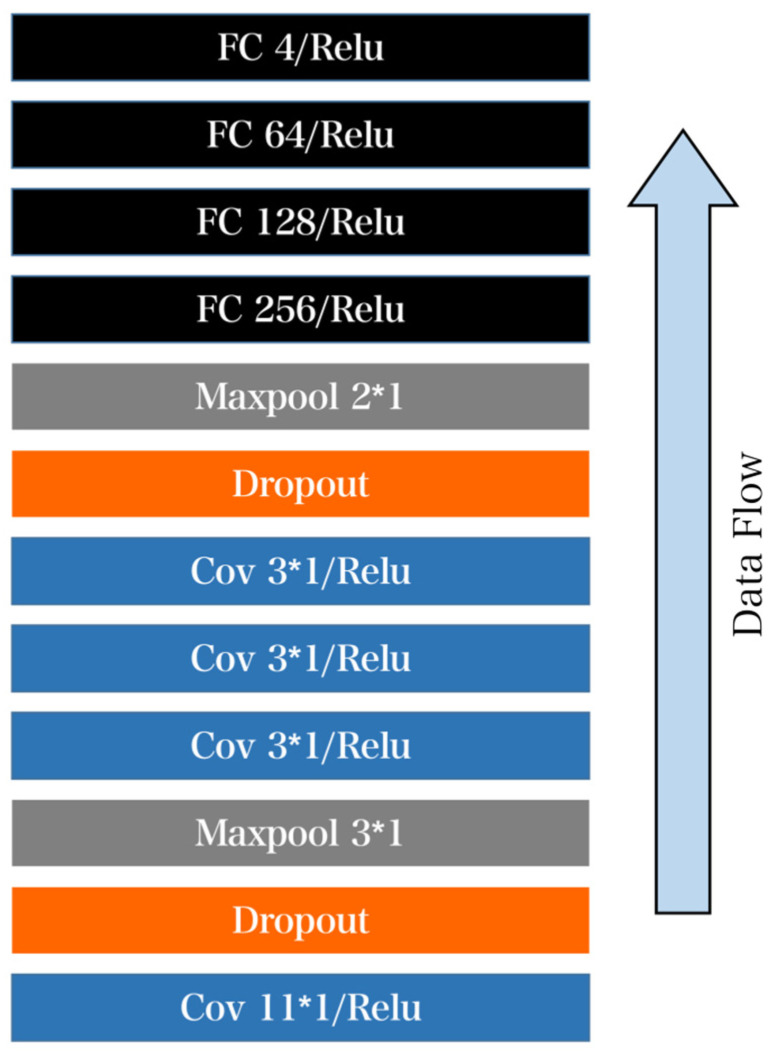
Diagram of CNN network structure. The core structure consists of four convolution layers and four fully connected layers. The size of the first convolution kernel is selected to be 11 × 1 and then the dropout layer is connected. ECG signals adjacent to each other have a great influence on information. In order to avoid information leakage, the size of 3 × 1 is selected for the three convolution cores after ECG signals.

**Figure 11 sensors-23-07643-f011:**
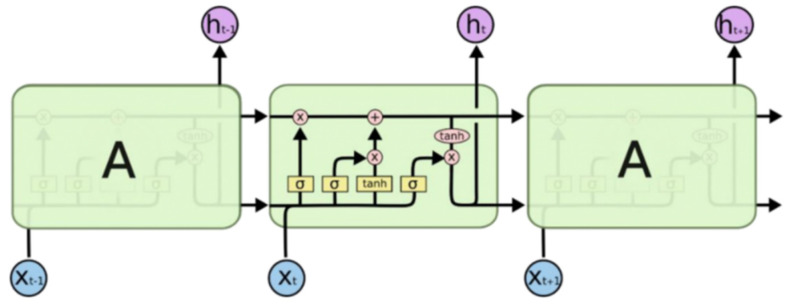
LSTM architecture diagram: the overall structure consists of a forget gate, input gate, and output gate [20].

**Figure 12 sensors-23-07643-f012:**
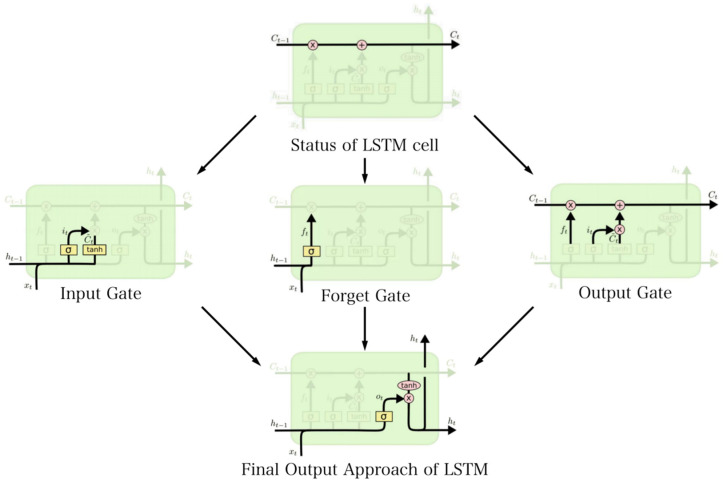
The structure of LSTM, including input gate, forget gate, and output gate. The advantage of LSTM compared with RNNs is that it performs an indiscriminate calculation on the entire sequence of information and selectively memorizes through gate control mechanisms.

**Figure 13 sensors-23-07643-f013:**
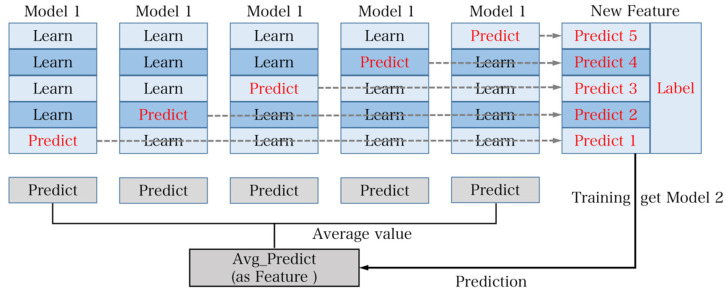
The calculation process for stacking. The core principle is layered training. The training set is divided into 5-fold cross-validation and prediction of the test set. The process is repeated 5 times and the data from these 5 repetitions are combined as new features and labels for the new training set. At the same time, the results of the 5 predicted test sets are averaged and utilized as new features for training Model 2.

**Figure 14 sensors-23-07643-f014:**
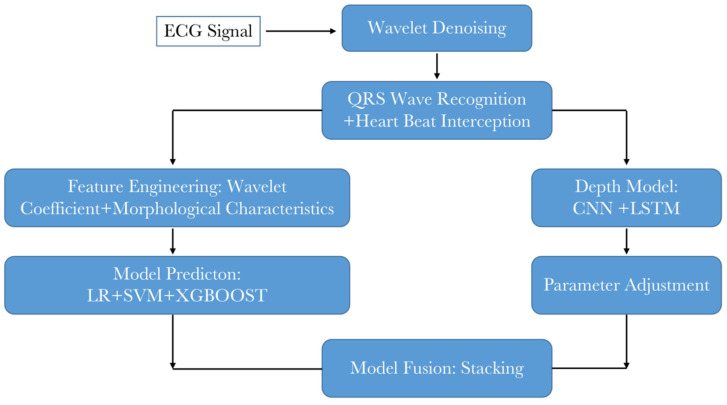
Algorithm diagram of the whole project.

**Table 1 sensors-23-07643-t001:** Classification Results of Different Types of Models.

Mark Type		LR Accuracy (%)	SVM Accuracy (%)
All		96.65	96.56
N	Normal pulsation	99.80	99.74
L	Left bundle branch block	98.73	98.00
R	Right bundle branch block	98.23	97.84
V	Ventricular premature beat	90.88	90.65

**Table 2 sensors-23-07643-t002:** Parameters of XGBOOST.

	Parameter Type	Parameter Value
Eta	Learning rate	0.3
Max_depth	Depth of decision tree	6
Lambda	L1 regular term	5
Alpha	L2 regular term	3
seed	Random number seed	2020
colsample_bytree	Proportion of random sampling columns	0.9
colsample_bylevel	Proportion of random sampling rows	0.9

**Table 3 sensors-23-07643-t003:** Parameters of CNN.

Parameter	Parameter Meaning	Parameter Value
Learning_rate	Learning rate	0.001
Epoch	Number of data iterations	100
Batch_size	The amount of data per training	64
Seed	Random number seed	2020
Dropout	Probability of random deactivation of dropout layer	0.5

**Table 4 sensors-23-07643-t004:** CNN, LSTM, and model fusion classification results [26].

Mark Type	XGBOOST Accuracy (%)	CNN Accuracy (%)	LSTM Accuracy (%)	Model Fusion Accuracy (%)
All	96.64	98.13	98.68	99.13
N	99.60	99.85	99.84	99.95
L	98.15	99.50	98.18	99.60
R	97.77	99.95	99.48	99.20
V	92.18	96.43	97.21	97.74

**Table 5 sensors-23-07643-t005:** AAMI standard classification of CNN model results.

Mark Type	Disease Type	Accuracy (%)	Actual Test Data	Accurately Predict Data
All		99.16	100,670	99,831
N	Regular pulse	99.86	90,081	89,951
S	Supraventricular premature beat	79.79	2781	2219
V	Ventricular ectopic heartbeat	97.99	7001	6866
F	Ventricular fusion	99.25	801	795

## Data Availability

Not applicable.

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
