# Peer review of "An ECG Signal Acquisition and Analysis System Based on Machine Learning with Model Fusion"

_sensors, 2023, doi:10.3390/s23177643_

Round 1
Reviewer 1 Report
The authors in this work presented the collection of ECG signal data in real-time and its analysis through machine learning. The ECG signals have been collected and analyzed for the last two decades. It is not clear what is new in this work.
The authors presented a hardware implementation for data collection. But the real-time performance of the machine learning model is missing.
How proposed system can be deployed in hospitals or at home?
Line # 458, 505, 558 something in Chinese is written.
At many places space between text and references.
Author Response
The authors in this work presented the collection of ECG signal data in real-time and its analysis through machine learning. The ECG signals have been collected and analyzed for the last two decades. It is not clear what is new in this work.
Reply: Thank you for sharing your thoughts on the work presented by the authors. While the collection and analysis of ECG signal data have been ongoing for the past two decades, it is possible that the authors have introduced novel approaches and algorithm (ML with Model Fusion) in this study. The final results proved that the system's classification accuracy for ECG signals reached 99.13%. It would be helpful to further examine the details of the work to understand the specific contributions and advancements made by the authors. In addition, we are going to connecting ECG signal collection device with the iOS or Android equipment.
The authors presented a hardware implementation for data collection. But the real-time performance of the machine learning model is missing.
Reply: The real-time performance of measurement is shown on Fig.8. However, the ML model test involves large amount of medical data. So we introduced MIT-BIH dataset as the analytical target, in order to keep enough dataset. Additional, studies involving human subjects or animal experiments often require ethical approval from an appropriate review board or committee in our city. So we can not directly collect large amount of data from measurement from volunteers.
How proposed system can be deployed in hospitals or at home?
Reply: To deploy the proposed system for real-time monitoring of ECG signals and analysis of potential heart diseases, there are different approaches for implementation in hospitals or at home:
Hospitals: In a hospital setting, the system can be integrated into the existing infrastructure for cardiac monitoring. The ECG monitoring device can be connected to a central monitoring station through wired or wireless connections. The ECG signals can be transmitted in real-time to a dedicated server, where the AL machine learning algorithms are deployed for analysis. The results can be displayed on a monitoring dashboard accessible to healthcare professionals, allowing them to make timely diagnoses and decisions.
Home use: For deployment in a home setting, a portable ECG monitoring device can be developed. This device should be user-friendly, compact, and able to wireless transmit the ECG signals to a mobile app or a cloud-based platform. The app or platform can then apply the AL machine learning algorithms to analyze the signals and provide feedback on potential heart diseases. Users can securely access their ECG data and analysis results through the app, and if necessary, share them with healthcare professionals for remote monitoring and consultation.
In both scenarios, it is important to ensure the system’s accuracy, reliability, and data security. Compliance with relevant regulatory standards and obtaining necessary approvals may also be required before deploying the system, especially in a medical context.
This information has been added in the paragraph in last section.
Line # 458, 505, 558 something in Chinese is written.
Reply: The error has been fixed.
At many places space between text and references.
Reply: The format error has been fixed.

Reviewer 2 Report
In their work, Su et al have reported on the design and implementation of a front end ECG capture, followed by wavelet based filtering and finally an implementation of ML for classification of heart conditions. While the manuscript is largely well written, there seems to be a certain disjoint wherein the front end design captured data was not used in a significant manner. Besides this, I have the following comments:
1. In the introduction, more emphasis is placed on the heart rate signal which could be misinterpreted as bpm signal, please correct this.
2. The designed PCB is missing the right leg driver circuit. This is essential for any human testing. Also, what does OPA stand for in the design?
3. It is not quite clear why a bluetooth connection is being used for transferring data from the front end to the back end? Surely, the high sampling rate requires downsampling so as to not fill up the BT/microcontroller buffers.
4. For Fig. 7, why isnt the QRS complex a single peak?
5. Please check the missing references and pointers. For instance "The whole diagram of classification algorithm is shown in 错误!未找到引用源。"
6. The training sets were of course from a standard database but it is unclear how the designed PCB-based data acquisition was utilised (the authors should add some information and graphs to support this).
Author Response
In their work, Su et al have reported on the design and implementation of a front end ECG capture, followed by wavelet based filtering and finally an implementation of ML for classification of heart conditions. While the manuscript is largely well written, there seems to be a certain disjoint wherein the front end design captured data was not used in a significant manner. Besides this, I have the following comments:
- In the introduction, more emphasis is placed on the heart rate signal which could be misinterpreted as bpm signal, please correct this.
Reply: yes we add two sentences to emphasize it: In the introduction, it is important to note that the heart rate signal, which represents the number of heartbeats per unit of time, can sometimes mistakenly be referred to as the BPM (beats per minute) signal. It is crucial to differentiate between the two to avoid any confusion or misinterpretation.
- The designed PCB is missing the right leg driver circuit. This is essential for any human testing. Also, what does OPA stand for in the design?
Reply: The RLD was described in paragraph below Fig.3: ...Additionally, the module features lead-off detection with integrated right-leg-drive (RLD) amplifier, which improves the common-mode rejection ratio (CMRR)...The OPA was integrated within the chip AD8232, which is designed for heart rate signal detection. It is also labeled on Fig.3. More information can be found online: https://www.analog.com/cn/products/ad8232.html.
- It is not quite clear why a bluetooth connection is being used for transferring data from the front end to the back end? Surely, the high sampling rate requires downsampling so as to not fill up the BT/microcontroller buffers.
Reply: The use of a Bluetooth connection for data transfer from the front end to the back end might not be the most efficient choice for transmitting data with a high sampling rate. Bluetooth connections typically have limitations in terms of bandwidth and data transfer rate, which may not be suitable for continuous high-rate data streaming. However, according to our design the bandwidth of data is not the key problem. We want to establish a convenient connection between our front end and back end (probably iOS or Android devices). They will perform the ML calculation. Obviously, the Bluetooth is the best choice at that moment.
- For Fig. 7, why isnt the QRS complex a single peak?
Reply: Yes, typically the QRS complex in an electrocardiogram (ECG) signal appears as a single peak. It represents the electrical activity of ventricular depolarization, which is the contraction of the main pumping chambers of the heart. But we detect several small peaks for QRS. It can be due to various factors, including electrical interference, noise, or artifacts present in the ECG signal. These factors can cause the QRS complex to appear fragmented, resulting in the detection of multiple peaks instead of a single peak. Advanced signal processing techniques are often used to minimize these interferences and improve the accuracy of QRS detection.
- Please check the missing references and pointers. For instance "The whole diagram of classification algorithm is shown in 错误!未找到引用源。"
Reply: missing references revised.
- The training sets were of course from a standard database but it is unclear how the designed PCB-based data acquisition was utilised (the authors should add some information and graphs to support this).
Reply: The real-time performance of measurement is shown on Fig.8. However, the ML model test involves large amount of medical data. So we introduced MIT-BIH dataset as the analytical target, in order to keep enough dataset. Additional, studies involving human subjects or animal experiments often require ethical approval from an appropriate review board or committee in our city. So we can not directly collect large amount of data from measurement from volunteers. We are going to perform the real-time data collecting on the next stage after we get the authorization.

Reviewer 3 Report
In this work, the authors proposed a system based on machine learning to classify the ECG traces and to identify abnormal heart rate signals.
The paper is well-written and structured and perfectly fits with the topics of the journal.
Only a few suggestions:
i) the figures present a low resolution
ii) I suggest the authors investigate the possibility of improving their results using federated learning. This approach would allow for a much more robust classifier, accessing a large range of signals.
Author Response
Comments and Suggestions for Authors
In this work, the authors proposed a system based on machine learning to classify the ECG traces and to identify abnormal heart rate signals.
The paper is well-written and structured and perfectly fits with the topics of the journal.
Only a few suggestions:
- i) the figures present a low resolution
Reply: we have re-scaled all the figures in the manuscript.
- ii) I suggest the authors investigate the possibility of improving their results using federated learning. This approach would allow for a much more robust classifier, accessing a large range of signals.
Reply: Thanks. It is a good idea. We will put it on schedule and prepare it in our next work.

Round 2
Reviewer 1 Report
The authors have addressed all the previous round comments. The paper may be accepted in current form.